# Construction and Research Progress of Animal Models and Mouse Adapted Strains of Seasonal Influenza Virus

**DOI:** 10.3390/vaccines13101077

**Published:** 2025-10-21

**Authors:** Haijun Zhu, Siyu Pu, Peiqing He, Junhao Luo, Rongbao Gao

**Affiliations:** 1Department of Health Inspection and Quarantine, School of Public Health, Public Health College, Anhui Medical University, Hefei 230032, China; 2NHC Key Laboratory of Biosafety, NHC Key Laboratory of Medical Virology and Viral Diseases, Chinese National Influenza Center, National Institute for Viral Disease Control and Prevention, Chinese Center for Disease Control and Prevention, Beijing 102206, China

**Keywords:** influenza, mouse adaptation, viral pathogenesis, reverse genetics, polymerase complex, vaccine evaluation

## Abstract

Influenza viruses, featured by high variability, pose a persistent public health threat because of an annual seasonal epidemic in the world and irregular global pandemic, requiring animal models to elucidate their pathogenic mechanisms and advance preventive strategies. Mice have been selected as the primary animal model, although several experimental animals have been used in studies of the influenza virus. However, the limited susceptibility of wild-type influenza viruses to mice poses significant challenges for studying pathogenesis and intervention strategies. Here, to help understand the construction of mouse-adapted influenza viruses, we reviewed the recent research progress in constructing mouse-adapted influenza virus strains to overcome species-specific barriers.

## 1. Introduction

Influenza viruses are enveloped, negative-sense RNA viruses assigned to the viruses infecting humans cause the family Orthomyxoviridae. Each winter, they seed acute respiratory outbreaks, and their scale oscillates between seasonal epidemics, responsible for substantial morbidity and billions in healthcare costs, and, historically, global pandemics, which resulted in the millions or even tens of millions of deaths [1,2]. WHO estimates annual seasonal influenza at roughly one billion infections, three to five million of them progress to severe disease, and 290,000–650,000 to fatal outcomes in the world [3]. To conduct in-depth research on the pathogenesis or prevention, and strategies for controlling influenza viruses, especially for those that pose a greater threat to humans, it is crucial to establish suitable animal models. However, most of the influenza viruses, including seasonal influenza viruses, cannot infect mice naturally because of their species selectivity, which could be influenced by several factors, including receptor binding specificity, polymerase activity differences, and/or host factor dependence [4]. To overcome these species barriers, researchers have engineered a suite of mouse-adapted strains in humanized transgenic mice or in ex vivo lung-culture systems. These complementary models now permit dissection of both influenza A (IAV) and B (IBV) virus pathogenesis and allow pre-clinical screening of next-generation vaccines or antivirals under controlled and reproducible conditions. In addition, constructing mouse-adapted strains can track the adaptive changes of the influenza virus in heterologous hosts, including gene mutations, reassortments, etc. It also helps to understand the evolutionary laws and adaptive mechanisms of the virus, to predict the epidemic trend and potential threats of the virus, and to provide early warning and decision-making for public health prevention and control. In this review, to enhance understanding of the mouse-adapted strains of influenza viruses, we summarized research progress on the design principles, key adapted mutations, and mechanistic insights of mouse-adapted influenza strains construction, commented on their contributions and limitations in basic or applied studies, and explored their challenges in the future.

## 2. Influenza Virus

Based on the antigenic divergence of their nucleoprotein (NP) and matrix (M) proteins, influenza viruses are classified into four distinct types: A, B, C, and D [5,6,7]. Among these pathogens, Influenza A Virus (IAV) can be further classified into multiple subtypes based on the antigenic differences of surface proteins, namely hemagglutinin (HA) and neuraminidase (NA). Currently, 18 types of HA and 11 types of NA have been identified, and their combinations give rise to subtypes such as H1N1, H3N2, H5N1, etc. IAV has a broad host range, capable of infecting various animals, including humans, birds, pigs, and horses. It is one of the primary pathogens responsible for human influenza and a major cause of global pandemics. Birds, particularly wild waterfowl, serve as one of the natural reservoirs for Influenza A Virus (IAV). Multiple subtypes of IAV can be carried in the intestines of birds over extended periods and are transmitted to other avian species as well as mammals. In contrast, Influenza B Virus (IBV) primarily infects humans and is rarely found in non-human animals, with only a few cases reported in seals and other limited species. Nonetheless, IBV is still capable of causing seasonal influenza epidemics in humans, often leading to severe illness, particularly among children and young adults. Since it was first reported in 1940, Influenza B Virus (IBV) has gradually evolved into two separate lineages—namely, the B/Victoria/2/87-like lineage and the B/Yamagata/16/88-like lineage. The two lineage viruses can be further divided into different antigenic groups. Influenza C virus (ICV) is a widely prevalent pathogen among children, typically causing only mild respiratory symptoms [8,9]. In addition, ICV can infect various animals, including pigs, dogs, and cattle [10]. However, ICV is unlikely to pose a major threat to human health [11,12]. Influenza D virus (IDV) is quite rare, and there is no record of human infection cases. IDV mainly infects a limited range of animals, including pigs and cows, with minimal impact on public health [13].

Seasonal influenza is caused by known Influenza A (H1N1 and H3N2) and B viruses, with annual epidemics driven by antigenic drift. It primarily affects infants, the elderly, and individuals with chronic medical conditions, with a case fatality rate typically below 0.1% [14]. Influenza pandemics are triggered by novel subtypes of Influenza A virus (such as those originating from cross-species transmission of avian or swine viruses), that target preferentially in subjects who typically lacks immunity, and lead to rapid global spread, high rates of severe illness among young adults (as seen with the 1918 H1N1 pandemic) and significantly higher case fatality rates (up to 5%) [15]. Pandemic influenza lacks a predictable seasonal pattern and may co-circulate with seasonal influenza viruses within communities. Historically, after the emergence of a novel influenza virus that triggers a pandemic, the then-circulating seasonal influenza strains may disappear [16]. Although seasonal influenza and influenza pandemics differ in terms of transmission scope and severity, they share similarities in their pathogenic characteristics. Both primarily induce inflammatory responses by infecting respiratory epithelial cells, leading to typical symptoms in patients such as high fever, muscle aches, fatigue, dry cough, and sore throat. The pathogenesis is closely related to the virulence and genetic characteristics of the virus, as well as the host’s immune status. Seasonal influenza generally causes milder illness, with most patients recovering spontaneously, while high-risk populations are prone to complications such as pneumonia and myocarditis. In contrast, during influenza pandemics, due to the widespread lack of immunity in the population, the number of infections and severe cases is much higher, symptoms may be more severe, and both morbidity and mortality rates are significantly elevated compared to seasonal influenza [14,15]. The prevention and control measures for seasonal influenza or influenza pandemics mainly include vaccination, antiviral drug therapy, and non-pharmaceutical interventions. Vaccination stands as a pivotal strategy, yet it faces challenges, including the readiness of vaccines due to viral mutations or insufficient vaccine supplies during the early stages of a pandemic. Antiviral drugs can effectively treat and prevent influenza, but there are several limitations, including the timing of medication and drug resistance issues. Non-pharmaceutical interventions, such as personal hygiene practices and isolation measures, can slow down transmission, but their implementation requires balancing socioeconomic impacts. Additionally, during an influenza pandemic, medical resources may be overwhelmed, and the public’s awareness and compliance with prevention measures need improvement, all of which pose significant challenges to influenza control efforts [17].

## 3. Animal Models for Influenza Virus Research

### 3.1. Mouse Model

Mice are widely utilized in fundamental research on influenza viruses due to their well-defined genetic background, short reproductive cycle, and low cost. However, wild-type Influenza A viruses typically struggle to replicate efficiently in the respiratory tracts of mice, owing to the presence of an interferon-induced restriction factor called Mx1 in inbred mouse strains [18]. Meanwhile, there are differences in the distribution of sialic acid receptors on the surface of respiratory epithelial cells between humans and mice: humans predominantly express α2,6-sialic acid receptors, whereas the mouse respiratory tract mainly features α2,3 receptors [19]. This discrepancy restricts the direct application of wild-type clinical isolates in mouse models, making the construction of mouse-adapted strains a crucial breakthrough. Adapted strains obtained through serial passaging or genetic engineering modifications not only mimic the infection characteristics of the virus infection in the human body but also are used to uncover the molecular mechanisms underlying host-specific adaptation. The primary advantage of using mice in influenza virus research is that their lung pathology closely resembles human cases of viral pneumonia [20]. The adapted influenza viruses can induce primary viral pneumonia in mice, mirroring the condition in humans, and leading to low blood oxygen saturation and elevated lactate dehydrogenase levels, which can serve as indicators for monitoring disease progression. Pathological damage in lung tissues can be a key criterion for assessing viral pathogenicity. Additionally, weight loss and survival rates in mice are reliable markers of the severity of influenza disease. Unlike humans, mice infected with influenza viruses do not experience fever but instead show a drop in body temperature. Symptoms such as cyanosis, dyspnea, and hemoptysis are also not readily observable in mice. Furthermore, the pathogenicity of influenza viruses varies among different mouse strains, for instance, compared to BALB/c and C57BL/6 mice, DBA/2 mice infected with PR/8 mice exhibit greater susceptibility, more rapid weight loss and mortality, higher cytokine production, and more severe lung tissue pathology [21].

Although the pathogenesis of the influenza virus in humans has not been fully simulated in mice, the convenience of monitoring clinical symptoms (including weight loss) has driven their extensive use in preclinical studies for evaluating the efficacy of many influenza vaccines and antiviral drugs. Additionally, the commercial availability of mouse-specific reagents facilitates research on immune responses to influenza virus infection. Furthermore, the use of gene knockout or deficient mouse models has also promoted their applications in the field of influenza research. However, despite these notable advantages, the use of mouse models in biomedical research on influenza virus infection still faces certain limitations. Apart from highly pathogenic strains such as H1N1pdm09, H5N1, H7N7, and H7N9, mice do not naturally obtain infections by influenza viruses [22]. Additionally, the practicality of mouse models in studying influenza virus transmission is limited. The vast majority of primary human influenza virus isolates are not infectious or pathogenic in BALB/c or C57BL/6 inbred mouse strains. Only under specific conditions or in particular mouse strains, it becomes possible for the viruses to infect and cause disease in mice. Transmission among mice can occur only after infection under specific conditions, in particular, mouse strains or with specific virus isolates. For instance, the recently reported humanized DRAGA mice have been used as a transmission model for Influenza A virus infection, as well as the H2N2 mouse-adapted strain reported in the 1960s [22,23,24].

### 3.2. Ferret Model

Since ferrets were first utilized as an animal model for influenza infection in 1933, they have been widely applied in influenza research, encompassing the evaluation of viral pathogenicity, transmissibility, viral tropism, host immune responses, as well as the development of novel vaccines and antiviral therapies. Ferrets serve as an experimental animal model for studying influenza virus infections due to their innate susceptibility to both Influenza A and B viruses. Moreover, their clinical manifestations and pathological changes associate with bronchitis and pneumonia and closely resemble those observed in humans [25,26,27,28]. Ferrets share similar lung physiology and cellular receptor distribution with humans and are highly susceptible to a wide range of influenza virus strains, including both Influenza A and B viruses. Similar to humans, seasonal influenza viruses primarily infect the upper respiratory tract tissues of ferrets, while highly pathogenic strains can invade the lower respiratory tract. Upon infection, ferrets exhibit clinical symptoms akin to those seen in humans, encompassing fever, nasal discharge, coughing, gastrointestinal complications, serum abnormalities, neurological complications, weight loss, anorexia, lymphopenia, and/or lethargy. Unlike mouse infection models, ferrets serve as an ideal animal model for studying influenza virus transmission. The virus can spread among ferrets through both direct and indirect contact (via aerosols, respiratory droplets, or airborne transmission). Furthermore, the airborne transmissibility of influenza viruses among ferrets is positively correlated with the quantity of infectious particles exhaled by the infected (virus donor) ferrets.

Given that ferrets exhibit disease manifestations similar to those of human influenza, their immune response to influenza virus infection is likely more relevant to humans than that of mice. However, the use of ferret models is constrained by several factors, including the lack of specific commercial reagents, their relatively large size, and the high costs and stringent requirements associated with their maintenance. Additionally, due to the transmissible nature of influenza viruses in ferrets, there is a risk of virus leakage when conducting research on highly pathogenic influenza strains in this model. Consequently, such work should be carried out in high-level biosafety laboratories housed in appropriate facilities [29].

### 3.3. Guinea Pig Model

Guinea pigs have been used as a model for influenza virus transmission [30]. Guinea pigs possess lung anatomical structures and physiological characteristics similar to those of humans. Since their first application in influenza virus research in 1963, guinea pigs have been relatively widely used in this field due to their small size, ease of procurement, and relatively low cost. Guinea pigs are naturally susceptible to various influenza virus strains without prior adaptation. However, the symptoms following infection are mild and not easily observable, and the infection does not lead to death in guinea pigs. Even when H5N1 and the 1918 pandemic virus replicate in the lungs and nasal turbinates of these animals, no weight loss or morbidity is observed [31]. After influenza infection, the virus is primarily confined to replicating in the cells of the upper respiratory tract tissues of guinea pigs, though it may also invade the lungs. However, the level of viral replication in lung tissues is typically significantly lower than that in the nasopharynx. Similar to ferrets, guinea pigs are also widely employed to evaluate the transmission potential of influenza viruses. They can contract the virus through aerosolized contaminants, minute particles released from virus-contaminated surfaces, or respiratory droplets. Additionally, guinea pigs are utilized in studies assessing the protective efficacy of vaccines. Nevertheless, the mild symptoms exhibited by guinea pigs following influenza virus infection make them less suitable for investigating the pathogenesis of influenza. Furthermore, the lack of specific commercial reagents also restricts their broader application in this field [29].

### 3.4. Hamster Model

The Syrian hamster is another animal model that can be used for the pathogenesis, transmission, and vaccine research of the influenza virus. The establishment of this model is based on the susceptibility of Syrian hamsters to the human influenza virus H3N2 and various avian influenza viruses [32]. In addition, because the body temperature of hamsters is close to human, it is considered to be an important influenza vaccine evaluation model. Another report showed that pandemic H1N1 and seasonal H3N2 influenza virus isolates could be airborne among Syrian hamsters [22]. However, the hamster model has not been widely used in influenza research because it lacks clinical symptoms even if the respiratory virus titer is high after infection. In addition, the lack of commercial reagents for immunological assays also limits their application.

### 3.5. Cotton Rat Model

Cotton rats are well-established models for respiratory syncytial virus (RSV) and are also considered a model for human influenza viruses. Similar to ferrets and guinea pigs, cotton rats can be infected with human Influenza A or B viruses without prior adaptation [33]. However, to date, no data have been reported regarding the ability of influenza viruses to transmit among cotton rats. Following intranasal inoculation with a high dose of influenza virus, cotton rats presented various pathogenic manifestations, including hypothermia on days 1, 2, and 10 post-infection, up to a 90% reduction in body weight, and a 173% increase in respiratory rate (tachypnea) [34]. These symptoms are dose-dependent. Histopathological analysis of influenza virus-infected cotton rats reveals lesions primarily localized in the lower respiratory tract. While nasal inflammation was not observed following H3N2 influenza virus infection, there was evidence of columnar epithelial cell shedding in the larger pulmonary airways, along with pronounced interstitial and alveolar pneumonia.

### 3.6. Pig Model

Due to notable similarities with humans in terms of genomic sequences, anatomy, and physiology, pigs serve as a crucial animal model for studying the pathogenesis and transmission of influenza, as well as for evaluating influenza vaccines. They can be utilized to investigate various aspects of influenza viruses, including infection, transmission, pathogenic mechanisms, and the efficacy assessment of vaccines or drugs. Pigs hold significant importance for research on cross-species transmission and the emergence of novel influenza viruses. As natural hosts for Influenza A viruses, pigs possess both SA α-2,6 in their upper respiratory tracts and SA α-2,3 in their lower respiratory tracts, making them ideal hosts and “mixing vessels” for the infection and recombination of avian and human influenza viruses. Pigs present influenza-like symptoms such as fever, cough, loss of appetite, and dyspnea after infection with both Influenza A and B viruses. In addition, these infections may also induce pulmonary lesions [35]. The pig model is well-suited for evaluating the efficacy of influenza vaccines, as, similar to observations in humans, the level of HA-specific neutralizing antibodies following influenza infection in pigs correlated with protection against swine influenza [35,36]. Additionally, the pig model has been employed in studies assessing vaccine-associated enhanced respiratory disease. However, the high cost of pig rearing, the relative complexity of experimental procedures, and the potential inadequacy of the pig infection model for certain human-specific influenza virus strains limit its widespread application [37]. Miniature pigs and conventional pigs show no significant differences in genomic sequences, receptor distribution, and susceptibility to Influenza A viruses, making miniature pigs a promising alternative to conventional pigs in influenza research.

### 3.7. Non-Human Primate Model

Non-human primates (NHPs) share close genetics with humans, coupled with similarities in anatomy, physiology, and immune characteristics, making them an invaluable animal model for studying influenza viruses. Currently, species used in influenza research include rhesus macaques, cynomolgus macaques, pig-tailed macaques, and African green monkeys, all of which are sensitive to many Influenza A isolates without prior adaptation. In 2013, the common marmoset was also demonstrated to be susceptible to human influenza virus isolates, with the virus capable of transmitting between donors and recipients [38]. After infection with influenza viruses, NHPs presented clinical symptoms such as conjunctivitis, lethargy, anorexia, and nasal discharge, which are highly consistent with human manifestations after infection. Consequently, NHPs have been widely employed by numerous laboratories to investigate the pathogenesis of influenza viruses and to evaluate the immunogenicity and efficacy of novel influenza vaccine candidates or therapeutic approaches. However, the use of NHPs in experiments raises ethical concerns, imposes high costs, requires complex operational procedures, necessitates specialized housing environments, and demands handling by experienced personnel. Additionally, restrictions on the number of NHPs that can be used further limit the widespread application of this model [39].

### 3.8. Chicken (Chick)

Although chicks serve as natural hosts for some influenza viruses, featuring α-2,3 sialic acid receptors and low feeding costs, making them suitable for poultry-related research, their physiological structures and immune systems differ significantly from those of humans. Consequently, the predominantly expressed α-2,3 receptors do not align with the α-2,6 receptors in the human respiratory tract, leading to weak replication and pathogenicity of H3N2 in chicks. This makes it challenging to simulate infection characteristics and immune responses in humans, rendering chicks unsuitable for simulating human infections and evaluating vaccine efficacy.

## 4. Construction Method of Influenza Virus Mouse-Adapted Strain

### 4.1. Traditional Passaging Adaptation Method

Most influenza viruses can adapt to mice through continuous lung-to-lung passage, acquiring the ability to infect, replicate, and cause pathological changes in the respiratory tissues of mice. The basic process is illustrated in Figure 1. During the process, the viral genome gradually undergoes adaptive mutations under the pressure of the host’s immune response, enabling the virus to acquire certain functions that play a crucial role in the acquisition of virulence by the viral strain. This, in turn, increases viral titers in the lungs and enhances pathogenicity, potentially leading to morbidity or even mortality in mice [40,41,42]. During the process of obtaining mouse-adapted viral strains, in addition to serial passage within the same mouse strain, parallel and cross-passage can also be conducted simultaneously across multiple mouse strains, as well as passage in chicken embryos, as shown in Figure 1. By performing parallel and cross-passage among multiple mouse strains, the virus is exposed to a broader range of host environments, facilitating the generation of more diverse mutations. This approach aids in the selection of viral strains with enhanced adaptability and broader infectivity. Moreover, it simulates natural selection pressures, thereby avoiding the potential oversight of adaptive viral strains due to the limitations of a single mouse strain. Additionally, passage in chicken embryos serves as an effective tool for the amplification and purification of influenza viruses, enabling the acquisition of sufficient viral quantities for subsequent experiments while preserving the original characteristics of the viruses, especially for strains directly isolated from natural hosts. This facilitates the study of mechanisms underlying cross-species transmission of viruses. These methods collectively enhance the efficiency and adaptability of obtaining mouse-adapted viral strains [43,44]. To use contemporary seasonal A/H3N2 viruses, mouse-adapted A/H3N2 influenza viruses were generated by continuous lung-to-lung blind passage in immunosuppressed mice, followed by additional lung-to-lung blind passage in immunocompetent (IC) mice [45]. In a study evaluating vaccine efficacy, an adaptive passage approach was employed to establish a viral strain capable of replicating efficiently in both mice and chicken embryos for the purpose of vaccine efficacy assessment. The passage process initially involved serial passage of the original virus, A/Indiana/08/2011 (H3N2v), seven times in eight-week-old female BALB/c mice to select for viruses with enhanced replicative capacity in the lungs. Subsequently, the virus underwent three additional passages in chicken embryos to increase its yield in this system. This strategy successfully constructed an adapted strain with heightened virulence [46,47]. In another mouse adaptive strain construction experiment, researchers achieved the gradual adaptation of the virus to the host immune system by conducting passages in mice of different ages. The first lung-to-lung passage was performed on four-week-old female balb/c mice for six generations, then the lung-to-lung passage was continued on five-week-old female balb/c mice for seven generations, and five generations were continued on six-week-old female balb/c mice, and finally, a highly pathogenic adaptive strain was obtained in the eighteenth passage [48]. This strategy not only mimicked the evolutionary process of the virus in its natural host but also enhanced the reliability and applicability of the model, laying a foundation for subsequent research on pathogenic mechanisms and vaccine evaluation. In another study, to overcome host restrictions, enhance pathogenicity, and simulate natural evolution, mouse-adapted H3N2 viruses were generated through continuous lung-egg-lung passage in Balb/c and DBA/1J mice, resulting in higher mortality rates in the mice. Subsequently, it was confirmed that the HA changes in the mouse-adapted viruses were the primary factor contributing to the increased pathogenicity, and mutations in NP and PB2 also contributed to cross-species adaptability [49]. The aforementioned passage numbers ranged from 11 to 18, with mouse ages spanning from young age (4-week-old) to adult (8-week-old) mice. From these passage schemes, it is evident that lung-egg-lung passages were initially conducted in mice with relatively compromised immune functions or those prone to infection, such as DBA/1J mice, immunosuppressed mice (IS), and young mice. Subsequent passages were carried out in immunocompetent or adult mice, with the option to incorporate chicken embryo passages to simulate the natural evolutionary process of the H3N2 virus. Exploring its mutation spectrum under these conditions holds greater practical significance.

### 4.2. Genetic Engineering Transformation Method

Based on known mutation sites in the virus that are associated with mouse-adaptation, recombinant viruses containing these mutations can be constructed using reverse genetics and other technologies. The basic process is shown in Figure 1. Studies have found that certain mutations in the PB2 gene of influenza viruses, such as E627K and D701N, are related to viral adaptability in mammals. Therefore, mouse-adapted strains can be obtained by constructing recombinant influenza viruses containing these mutation sites [45]. Genetic engineering methods offer numerous advantages, including the precise introduction of specific mutations, enabling modified viruses to rapidly adapt to mouse hosts. Compared to traditional approaches, these methods allow for more efficient acquisition of viruses with specific adaptive traits, shortening the time required for conventional adaptation processes and enhancing the success rate of adaptation. Additionally, they can, to a certain extent, reduce blindness and randomness, thereby improving the reproducibility of mouse-adapted viral strains. However, this method also has certain drawbacks. It requires specific molecular biology techniques and equipment, imposing high technical demands on the experimental personnel. During the operation process, strict adherence to relevant experimental norms and safety protocols is essential to prevent potential biosafety issues arising from uncontrolled gene mutations or recombination. Moreover, the process of constructing recombinant viruses is relatively complex, and there may be instances of failed construction or viruses with characteristics that do not meet expectations. Therefore, even after obtaining recombinant viruses, it is still necessary to conduct a certain number of passage experiments in mice to verify whether the expected adaptability has been achieved.

## 5. Molecular Mechanisms During Adaptation

Many adaptive gene mutations that viruses acquired through adaptive evolution under specific environmental conditions were observed as the result of the entire genome functional coordinated system. However, the recombinant viruses with these adaptive mutations may fail to recapitulate the phenotypes of the adapted strains when identified key mutations were individually introduced into the original viral backbone through reverse genetics technology. Those key mutations may rely on the functional support provided by others, such as yet undiscovered mutations within the genome or gene backbone. In the absence of this necessary genetic context, the target mutations not only fail to exert positive effects but may even lead to a decline in adaptability by disrupting the internal functional balance of the virus. Therefore, when elucidating the biological functions of specific mutations, it is essential to fully consider their genetic background and epistatic effects.

### 5.1. Molecular Mechanism of Effective Infection

#### 5.1.1. Polymerase Complex Mutations for Enhancing Viral Replication Efficiency

The influenza virus polymerase is a heterotrimeric protein consisting of three virus-encoded subunits of PB1, PB2, and PA. Adaptive mutations in these proteins may help overcome species barriers. The PB2 subunit is one of the core components of the influenza virus RNA polymerase complex. Its C-terminal domain (residues 538–759) contains a conserved nuclear localization signal (NLS) and a cap–binding domain, which is responsible for recognizing the 5′ cap structure of host mRNA and mediating the transcription initiation of viral RNA. The E627K mutation of PB2 is located in a highly conserved surface region, adjacent to the cap–binding domain (PDB: 4WRT). Structural analysis revealed that the E627K mutation introduced a positive charge, which may enhance the interaction between PB2 and host factors (such as importin-α) or RNA templates, thereby promoting the nuclear import and assembly efficiency of the ribonucleoprotein (RNP) complex [50]. Generally, the synthesis of viral genomic RNA (vRNA) by avian influenza virus polymerases is severely restricted in mammals, resulting in inefficient assembly of ribonucleoprotein complexes (RNPs) and insufficient viral protein expression. However, this limitation can be overcome through adaptive mutation repair mechanisms [51]. Most mammalian adaptive mutations of the influenza virus occurred in the PB2 protein, which is a key component of the viral polymerase to maintain high activity in mammalian cells [52,53]. Previous studies confirmed that PB2 residue site 271 plays a key role in enhancing the polymerase activity of Influenza A virus in mammalian host cells [54], the mutations of E627k and D710N are two well characterized substitutions in PB2 protein, which are essential for mammalian adaptation to a variety of avian influenza virus subtypes [54,55,56,57,58,59], the PB2 carrying the K526R mutation significantly enhances the activity of the viral polymerase complex particularly in mammalian cells, and K526R combined with known adaptive markers such as E627K or D701N present a synergistic effect on boosting the efficiency of viral RNA transcription and replication [60]. PB1 and PA have also been implicated to play a key role in mammalian adaptation besides lung virulence in mice [61,62,63,64,65,66,67,68]. Studies have shown that the V709I mutation combined with mutation V113A/K586R/D619N/V709I in PB1 protein of H3N2 mouse-adaptation strain significantly enhanced viral polymerase activity and early replication ability, while the mutation I709V in PB1 reduced viral activity in human cells, but showed a slight inhibitory effect in the mouse infection model, highlighting the role of host specific adaptation in virus evolution [69]. In addition, Research has demonstrated that PB2-D701N can enhance viral replication both in vitro and in vivo, and expand viral tissue tropism, and both the PB2-D701N mutation and the M1-M192V mutation are associated with the pathogenicity of mouse-adapted H3N2 avian influenza virus. A study on the adaptation of avian influenza viruses to mammals identified a key mutation (PB2-I714S) in H3N2 avian influenza viruses that significantly improved viral replication capacity and pathogenicity in mouse models by enhancing viral polymerase activity, promoting nuclear import efficiency, and optimizing RNP complex assembly. This key mutation has been considered to be a molecular mechanistic basis for constructing mouse-adapted H3N2 strains [70].

#### 5.1.2. Hemagglutinin (HA) and Neuraminidase (NA) Mutations for Changing Tissue Tropism

Substitution of amino acids in the hemagglutinin (HA) and neuraminidase (NA) of Influenza A virus is associated with host adaptation and increased virulence in mice [71,72]. The HA protein serves as a critical mediator for viral entry into host cells, with its structure comprising a head domain (receptor-binding domain, RBD) and a stalk domain (fusion domain). Mutations such as D190N and G186V, located near the receptor-binding site (RBS) of HA (PDB: 4O5N), may enhance HA binding affinity to α2,3-sialic acid receptors in the mouse respiratory tract by altering RBS conformation or charge distribution. Additionally, the T160A mutation, situated near a surface glycosylation site on the HA head, likely removes the glycan at this position, thereby increasing receptor accessibility while improving protein thermal stability and, consequently, viral infectivity in mice. Structural biology studies demonstrated that HA fusion activity depended on conformational changes in its stalk domain. Certain mutations (e.g., H311Y) promoted viral membrane–endosome membrane fusion by stabilizing the fusion intermediate state or lowering the pH threshold, thereby enhancing viral infectivity [73]. The NA protein facilitates viral particle release by cleaving sialic acid receptors, with its active site located in the head domain of a tetrameric structure (PDB: 1NN2). The S331R mutation in NA resides in a surface loop region, although not directly within the catalytic active center, and indirectly influences viral receptor binding and release kinetics by altering the functional balance between HA and NA. Structural modeling suggests that this mutation enhanced synergistic interactions between NA and HA, particularly following HA RBS mutations. The compensatory S331R mutation in NA helped to maintain proper viral particle aggregation and release, thereby optimizing viral transmission efficiency in the host [74].

Adaptive mutations in the HA protein may significantly enhance the adaptability of influenza viruses in mammalian hosts by altering receptor-binding specificity and optimizing protein stability and membrane fusion efficiency. These mutations not only strengthen the viral infectious capacity but also provide a molecular basis for cross-host transmission. The receptor-binding specificity of the HA protein is a critical factor for viral adaptation to different hosts. Common mutations in the HA receptor-binding domain, such as G186V and D190N, can modify the binding pattern of the HA protein to sialic acid receptors, making it more suitable for binding to α2,3-sialic acid receptors in mice. Certain HA mutations, such as T160A and I215T, increased the thermal and conformational stability of the HA protein, enabling HA to remain active in the high-temperature environment within mice. Continuous infection with influenza virus strains exhibiting different surface glycosylation patterns can trigger host immunopathology [75]. The membrane fusion function of the HA protein is a critical step for viral entry into host cells. Studies have found that certain mutations, such as H311Y, can increase the membrane fusion efficiency of the HA protein, thereby enhancing the virus’s infectious capacity [61,76,77,78,79,80,81]. The lack of a high-mannose oligosaccharide at HA residue 165 increased viral titers significantly in mouse lungs and resulted in lethal effects in mice [71]. In addition, mutations in the second sialic acid-binding site (2SBS) of the influenza virus neuraminidase (NA) can disrupt the functional balance between hemagglutinin (HA) and neuraminidase (HA–NA), thereby affecting the viral tissue tropism. This imbalance drives compensatory mutations in HA, which not only restore the HA–NA balance but may also alter the receptor-binding properties of HA by reducing its binding to avian α2,3-sialic acid receptors and potentially enhancing its binding to human α2,6-sialic acid receptors, ultimately increasing the viral potential to adapt to human hosts [82]. Mutations in the NA protein can act in synergy with hemagglutinin. For example, the NA-S331R mutation itself does not enhance virulence, but it provides a synergistic context for HA/polymerase mutations [49].

#### 5.1.3. Synergistic Mutation of Other Genes for Mice Adaptation

Genomic analysis of mouse-adapted strains derived from low-pathogenic avian influenza H3N2 viruses revealed mutations in the PB2 (E192K and D701N), PB1 (F269S, I475V, and L598P), HA (V242E), NA (G170R), and M1 (M192V) proteins, which synergistically play a role in enhancing the replication ability and pathogenicity of the H3N2 virus in mammals [83]. In another study, mutations combining PA (S184N) with PB2 (E627K) were identified to increase in pathogenicity of H3N2 canine influenza virus infections in mice or dogs, whereas the single mutation of PA (S184N) had no influence on viral pathogenicity in mice, demonstrating the synergistic role of the PA (S184N) and PB2 (E627K) mutations [48]. Nonetheless, no report has demonstrated that the development of virus adaptation can be simply determined by these mutations or synergistic mutations.

### 5.2. Mechanisms Causing Enhanced Pathogenicity

#### 5.2.1. Immune Escape

Non-structural protein 1 (NS1) plays an important role in blocking host innate cellular immune responses against the virus and inhibiting the host’s RNA interference pathway and protease activation, and its function may be critical for influenza virus pathogenicity [84,85]. The NS1 protein is a multifunctional immune evasion factor, with its structure composed of an N-terminal RNA–binding domain and a C-terminal effector domain. The E196K mutation, located in a surface β-sheet region of the effector domain and near the interaction interface with the host CPSF30 (cleavage and polyadenylation specificity factor 30), weakened the binding ability of NS1 to CPSF30, thereby reducing its inhibition of host mRNA processing and, in turn, enhancing the expression of viral RNA. Furthermore, a 7-amino acid deletion mutation at the C-terminus altered the conformational flexibility of NS1, affecting its interactions with TRIM25 or RIG-I, and thus weakening the host interferon response. Structural biology studies further revealed that the dimerization interface of NS1 was crucial for its function [86]. In another study, by using the A/Hong Kong/1/1968 (H3N2) (HK-WT) virus and incorporating mutated NS gene segments through reverse genetics, researchers demonstrated that all NS1 mutations were adaptive and enhanced viral replication in mouse cells and/or lungs (up to 100-fold). Another study revealed that the E196K mutation in the NS1 protein of H3N2 mouse-adapted strains significantly weakened its ability to suppress the host interferon response. Furthermore, the deletion of a 7-amino acid extension at the C-terminus further enhanced viral adaptability and virulence in the host through a synergistic effect [86]. In addition to NS1, influenza viruses also achieve immune evasion through the synergistic action of multiple proteins: PB1-F2 interacts with mitochondrial MAVS and IKKβ to suppress IFN production and NF-κB signaling; PB2 directly targets JAK1, blocking the JAK/STAT pathway; PA interferes with the nuclear translocation of IRF3, thereby inhibiting IFN-β transcription; NS2/NEP inhibits the nuclear transport of IRF7, limiting IFN expression; HA weakens type I interferon signaling by downregulating IFNAR1/IFNGR1 and inhibiting STAT1/STAT2 phosphorylation. Together, these mechanisms form a crucial immune evasion network that enables the virus to sustain replication and cause disease in mice [87].

#### 5.2.2. Inflammatory Storm and Organ Damage

“Inflammatory storm” refers to the excessive activation of the immune system after infection with the influenza virus, followed by sharp production of a large quantity of pro-inflammatory cytokines and chemokines in a short time period, which trigger a series of serious pathological reactions. After the influenza virus invades host cells, the viral RNA is recognized by pattern recognition receptors such as RIG-I, TLR3, and TLR7 in the cell, which initiate downstream signaling pathways to induce the production of type I and type III interferons, while activating the inflammasome and releasing pro-inflammatory cytokines such as IL-1β and IL-18. During the adaptation process in mice, the virus accumulates mutations such as PB2-E627K and HA-V242E through “gene–host” dual selection, enabling a rapid increase in its replication titer within the lungs. The sustained and massive replication of the virus triggers the overactivation of the NLRP3 inflammasome in alveolar epithelial cells and macrophages, leading to the cascading release of cytokines, including IL-1β, IL-18, TNF-α, IL-6, and CXCL1, which result in a “cytokine storm”. Excessive inflammatory mediators disrupt the alveolar-capillary barrier, causing pulmonary edema, neutrophil infiltration, and microthrombus formation, subsequently inducing hypoxemia. Meanwhile, cytokines disseminate through the circulation to the liver and kidneys, causing significant elevations in ALT, AST, BUN, and Cr levels, ultimately leading to multiple organ dysfunction. It is precisely the synergistic effect between efficient viral replication and excessive inflammatory responses that ultimately manifests as enhanced lethality in the mouse-adapted strain [21,88].

## 6. Conclusions and Prospects

The construction and application of mouse-adapted strains of influenza virus provide an irreplaceable research platform for the analysis of pathogenesis and the development of prevention and control strategies for influenza. Many scientific breakthroughs have been made to overcome the mouse host limitation of wild-type influenza viral strains and have revealed a series of key molecular adaptation mechanisms, such as HA receptor binding preference switching, polymerase activity enhancement, and intergenic synergy, through the strategy of combining traditional passaging screening with precise editing of reverse genetics. However, the construction of a mouse-adaptive strain still has some challenges, although the technology is relatively mature. Adaptive strains are not as easily obtainable as viruses based on reverse genetic technology yet. It would still require continuous optimization of the technical methods to improve the success rate of construction and the applicability of adaptive strains. Future research may make breakthroughs in the following aspects:

Technological innovation: To overcome the limitations of existing models, researchers have been exploring multiple innovative directions. Several approaches have been tested and can be further fine-tuned to precisely simulate the human infection microenvironment by integrating humanized receptor mouse models, respiratory organoids, and/or single-cell multi-omics technologies. The development of humanized mouse models with human influenza virus receptors would enable more accurate simulation of human infection and transmission processes.

Mechanism deepening: Utilize structural biology to elucidate the structure–activity relationship of mutant proteins and combine artificial intelligence to predict the pathways of adaptive evolution. The cryo-electron microscopy (cryo-EM) can resolve the specific structural changes induced by mouse-adapted mutations such as PB2-E627K and HA-V242E within the three-dimensional structures of the polymerase complex or hemagglutinin (HA). Subsequently, high-resolution structural data can be fed into the AlphaFold–ESM model, integrating in vivo deep mutational scanning to train graph neural networks. This approach enables the early prediction of cross-species adaptive mutations in viruses, providing a rapid design basis for updating vaccines and animal models. In AI-assisted virus evolution prediction, for instance, the E2VD model achieved a 67% improvement in accuracy for cross-virus prediction of “future beneficial mutations” and provided scoring for key sites such as PB2-E627K/HA-L194Q. Meanwhile, in the context of viral adaptation, CRISPR whole-genome screening combined with reverse genetics has confirmed that CMAS/B4GALNT2 and ADAR1 determine host adaptability by restricting PB2 activity [89].

Standardization System: To improve the reproducibility of research and the comparability of data, it may be crucial to establish standardized experimental protocols and a public adaptive strain database. Currently, significant variations exist among different laboratories in areas such as passaging strategies, mutation screening criteria, or phenotypic evaluation methods. It is difficult to compare research findings across studies. In the future, it would facilitate global research collaboration and accelerate progress in influenza virus research if standardized experimental procedures and data-sharing mechanisms were developed.

## Figures and Tables

**Figure 1 vaccines-13-01077-f001:**
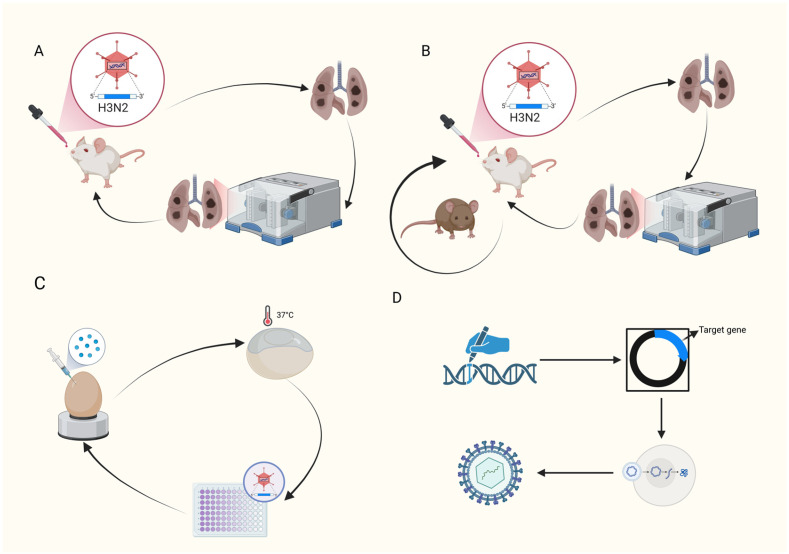
Basic process of virus passaging (Created in https://BioRender.com (accessed on 17 September 2025)). (**A**) Mice are inoculated with the virus through the nasal mucosa. After lesions develop, the diseased lungs are harvested, ground into a lung suspension, and this lung suspension is used for the next round of nasal mucosa inoculation in mice. (**B**) Mice are inoculated with the virus through the nasal mucosa. After lesions appear, the diseased lungs are collected, ground to form a lung suspension, which is then used for the next round of nasal mucosa inoculation in mice. After several generations of passage, mice of another strain are used for further cyclic passage. (**C**) The virus is injected into the allantoic cavity of chicken embryos using a syringe, followed by incubation in a 37 °C incubator. After a certain period, the allantoic fluid is collected for hemagglutination testing and then used for the next round of chicken embryo passage. (**D**) First, the target gene is identified and cloned into a viral vector plasmid. Subsequently, this recombinant viral vector plasmid, along with a helper plasmid system that provides trans-acting factors such as viral structural proteins, replicase, and envelope proteins, is co-transfected into packaging cells such as HEK-293T. Within the cells, the viral proteins expressed from the helper plasmids act in trans to package viral genomes expressed from the recombinant vectors into infectious recombinant viral particles. Finally, the supernatant is collected, purified, and the viral titer is determined to obtain a recombinant virus stock solution that can be used for subsequent experiments.

## Data Availability

The original contributions presented in the study are included in the article. Further inquiries can be directed to the corresponding author.

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
