# Peer review of "Construction and Research Progress of Animal Models and Mouse Adapted Strains of Seasonal Influenza Virus"

_vaccines, 2025, doi:10.3390/vaccines13101077_

Round 1

Reviewer 1 Report

Comments and Suggestions for Authors

This manuscript to review state-of-the-art animal models of influenza infections and construction of mouse-adapted influenza viruses is written properly for the field. However, the current manuscript is hardly readable, and extensive revision is required to provide correct English expression. The objectives and take-home messages of this Review also need to be explicitly stated. Details of the comments in red texts and/or red strikethrough are given in the attached pdf file.

Comments on the Quality of English Language

This manuscript requires extensive revision in English expression.

Author Response

Point 1: This manuscript to review state-of-the-art animal models of influenza infections and construction of mouse-adapted influenza viruses is written properly for the field. However, the current manuscript is hardly readable, and extensive revision is required to provide correct English expression. The objectives and take-home messages of this Review also need to be explicitly stated. Details of the comments in red texts and/or red strikethrough are given in the attached pdf file.

Response 1: Thank you so much for your comments and excellent suggestions. We reworded text throughout to make it clearer,and corrected the errors on English grammar after making double check in revised version.

Point 2: This manuscript requires extensive revision in English expression.

Response 2: Thank you for your important comment. We made double checks and corrected the errors on English grammar in revised version.

Reviewer 2 Report

Comments and Suggestions for Authors

Review comments on vaccines-3924740 

Manuscript details: 

Journal: Vaccines 

Manuscript ID: vaccines-3924740

Type of manuscript: Review 
Title: Construction and research progress of mouse adapted strains of seasonal influenza virus

Authors: Haijun Zhu, Siyu Pu, Peiqing He, Junhao Luo, Rongbao Gao *

Influenza Virus Vaccines

Major comments:

  • Overall, this review article does not have so significant impression.
  • In “Abstract” section, authors described about the methods they used for collecting literature as follows; “The literature of PubMed, web of science, andGoogle Scholar (2000-2025) was systematically analyzed using the keywords "seasonal influenza", "mouse adaptation", "host adaptation", and "influenza pathogenesis". Main adaptation strategies include continuous lung passaging of immunocompromised / young mice, cross species serial passaging, and reverse genetics targeting polymerase or/and hemagglutinin mutations”. However, this might be very unusual as a review article. If authors wanted to describe about such methods in this review article, they should appear in the main text; may be better to appear at the end part of introduction section.
  • The section on “animal models for influenza virus research” is well-written and concisely summarized. This part would be very useful for understanding the current situations about the animal models.
  • In section 5, authors tried to discuss about “the molecular mechanisms during the adaptation”. I feel that, in this section, discussions based on the 3D structural information on ribonucleoprotein complex (RNP), hemagglutinin (HA), neuraminidase (NA), and non-structural protein 1 (NSP1), respectively, are highly required. Without these pieces of information about their structural or mechanistic roles, arguments on the mutations would be somewhat tedious and meaningless.
  • As a whole, there are so many mistakes in the main text as listed below as Minor comments.
  • Accordingly, this review manuscript in the present form cannot be acceptable for publication in “Vaccines”.

Minor comments:

  • Reference number should appear just before a period in each sentence.
  • “morbility” (page 1, line 24) should be morbidity.
  • “Error! Reference source not found” (page 7, line 294 and 302; page 8, line 342; page 9, line 371) should be “Figure 1”.
  • About Table 1 (page 9), actually there is no citation on “Table 1” in the main text. Is this Table really necessary?
  • “Pb2” should be “PB2” (page 10, line 400).
  • “E627k”, “d710n”, and “Pb2” should be “E627K”, “D710N”, and “PB2”, respectively. (page 10, line 404).
  • “v709i” should be “V709I” (page 10, line 411).
  • “v113a/k586r/d619n/v709i” should be “V113A/K586R/D619N/V709I” (page 10, line 412).
  • “i709v” should be “I709V” (?) (page 10, line 413).
  • An unnecessary period at page 10, line 416.
  • “ofa” should be “of a” (page12, line 489).

Comments on the Quality of English Language

There are so many mistakes.

Author Response

Point 1: In “Abstract” section, authors described about the methods they used for collecting literature as follows; “The literature of PubMed, web of science, and Google Scholar (2000-2025) was systematically analyzed using the keywords "seasonal influenza", "mouse adaptation", "host adaptation", and "influenza pathogenesis". Main adaptation strategies include continuous lung passaging of immunocompromised / young mice, cross species serial passaging, and reverse genetics targeting polymerase or/and hemagglutinin mutations”. However, this might be very unusual as a review article. If authors wanted to describe about such methods in this review article, they should appear in the main text; may be better to appear at the end part of introduction section.

Response 1: Thank you for your comments. We are agreeing with your suggestion,and moved them to the main body for detailed description.

Point 2:In section 5, authors tried to discuss about “the molecular mechanisms during the adaptation”. I feel that, in this section, discussions based on the 3D structural information on ribonucleoprotein complex (RNP), hemagglutinin (HA), neuraminidase (NA), and non-structural protein 1 (NSP1), respectively, are highly required. Without these pieces of information about their structural or mechanistic roles, arguments on the mutations would be somewhat tedious and meaningles.

Response 2: Thank you for your suggestion. We've added the discussion based on the 3D structures of ribonucleoprotein complex (RNP), hemagglutinin (HA), neuraminidase (NA), and non - structural protein 1 (NS1) in section 5.

Point 3: As a whole, there are so many mistakes in the main text as listed below as Minor comments.

Reference number should appear just before a period in each sentence.

“morbility” (page 1, line 24) should be morbidity.

“Error! Reference source not found” (page 7, line 294 and 302; page 8, line 342; page 9, line 371) should be “Figure 1”.

About Table 1 (page 9), actually there is no citation on “Table 1” in the main text. Is this Table really necessary?

“Pb2” should be “PB2” (page 10, line 400).

“E627k”, “d710n”, and “Pb2” should be “E627K”, “D710N”, and “PB2”, respectively. (page 10, line 404).

“v709i” should be “V709I” (page 10, line 411).

“v113a/k586r/d619n/v709i” should be “V113A/K586R/D619N/V709I” (page 10, line 412).

“i709v” should be “I709V” (?) (page 10, line 413).

An unnecessary period at page 10, line 416.

“ofa” should be “of a” (page12, line 489)

Response 3: Thank you so much for your comments. We made double checks throughout of the text and corrected the mistakes in revised version. We corrected the issues on the reference number and all typing mistakes after making double check in the revised version, changed “morbility” to “morbidity” online 24 page 1, replaced “Error! Reference source not found” with “Figure 1” on line 294 and 302 page 7, line 342page 8 and line 371page 9. We removed the Table 1 after considering its necessity to the overall understanding of the content. We also removed the unnecessary period on line 416 page 10.

Reviewer 3 Report

Comments and Suggestions for Authors

The review paper presents a certain scientific interest, but there are some important comments:

  1. The title reflects the subject, but the abstract is brief and is not transparent in sentence linkage. Please revise the abstract and cover all review content.
  2. The review compares several animal models, including mice, ferrets, pigs, etc., without a criterion for comparison. Establish a uniform system (e.g., pathogenic similarity, immune response, transmission capacity, practicability) for model comparison.
  3. The overview mostly lists findings from earlier studies without discussing contradictions, limitations, or methodological heterogeneity critically. Add a comparative discussion of conflicting data (e.g., between mouse and ferret model results) and their implications.
  4. The adaptation mechanism and molecular mechanism sections are divided. Provide a bridging paragraph summary of how each construction method pertains to specific molecular mechanisms.
  5. While many references are up to date to 2025, some seminal papers on models of influenza adaptation (e.g., more recent reverse genetics applications and CRISPR-based host factor studies) are not cited. Provide more recent publications (2023–2025) on polymerase-host interaction, AI-assisted prediction of viral evolution, and organoid modelling.
  6. Some statements lack immediate citations (e.g., line 46–55 for host barriers). Provide in-text citations for mechanistic statements and at least one reference per major finding.
  7. Over-reliance on particular sources (e.g., Baz et al. 2019, Choi et al. 2020) may bias. Mix sources among geographic and methodological studies.
  8. Reduce sentence length and ensure a uniform scientific tone.
  9. There are some typographical, grammatical, and formatting errors. Please try to correct it.

Author Response

Point 1: The title reflects the subject, but the abstract is brief and is not transparent in sentence linkage. Please revise the abstract and cover all review content.

Response 1: Thank you for your comment. We revised the Abstract to make it clearer and cover all the review content.

Point 2: The review compares several animal models, including mice, ferrets, pigs, etc., without a criterion for comparison. Establish a uniform system (e.g., pathogenic similarity, immune response, transmission capacity, practicability) for model comparison.

 Response 2: Thank you for your feedback. We appreciate your suggestion to have a set way to compare animal models. However, we want to deliver the information about the advantages and disadvantages of available animal models for influenza viruses in this review. It would be better to read reviews or articles about animal models for influenza viruses if reader want to understand animal models for influenza virus deeply.

Point 3: The overview mostly lists findings from earlier studies without discussing contradictions, limitations, or methodological heterogeneity critically. Add a comparative discussion of conflicting data (e.g., between mouse and ferret model results) and their implications.

Response 3: Thank you for your feedback. We've added a comparison of conflicting data from different animal models (like mice and ferrets) in revised version.

Point 4: The adaptation mechanism and molecular mechanism sections are divided. Provide a bridging paragraph summary of how each construction method pertains to specific molecular mechanisms.

Response 4: Thank you for your suggestion. We've added a short summary in these sections to show how each construction method relates to specific molecular changes in revised version.

Point 5: While many references are up to date to 2025, some seminal papers on models of influenza adaptation (e.g., more recent reverse genetics applications and CRISPR-based host factor studies) are not cited. Provide more recent publications (2023–2025) on polymerase-host interaction, AI-assisted prediction of viral evolution, and organoid modelling.

Response 5: Thank you for your feedback. We cited newer papers (2023 - 2025) on polymerase - host interaction, AI - predicted virus evolution, organoid models and recent reverse genetics and CRISPR studies on influenza adaptation models in revised version.

Point 6: Some statements lack immediate citations (e.g., line 46–55 for host barriers). Provide in-text citations for mechanistic statements and at least one reference per major finding.

Response 6: Thank you for your feedback. We added in- text citations for all mechanism statements.

Point 7: Over-reliance on particular sources (e.g., Baz et al. 2019, Choi et al. 2020) may bias. Mix sources among geographic and methodological studies.

 Response 7: Thank you for your feedback. We mixed sources from different places and methods to give a more balanced view.

Point 8: Reduce sentence length and ensure a uniform scientific tone.

Response 8: Thank you for your feedback. We made double checks and reword them in revised version.

Point 9: There are some typographical, grammatical, and formatting errors. Please try to correct it.

Response 9: Thank you for your feedback. We made double checks and corrected the errors on English grammar in revised version.

Round 2

Reviewer 2 Report

Comments and Suggestions for Authors

Review comments on vaccines-3924740-revised version 

Manuscript details: 

Journal: Vaccines 

Manuscript ID: vaccines-3924740

Type of manuscript: Review 
Title: Construction and research progress of animal models and mouse adapted strains of seasonal influenza virus

Authors: Haijun Zhu, Siyu Pu, Peiqing He, Junhao Luo, Rongbao Gao *

Influenza Virus Vaccines

Major comments:

In the revised version, authors made appropriate revisions. Details and my comments are summarized below. Accordingly, this revised manuscript is now suitable for publication in “Vaccines” in the present form (after very minor revision indicated below in minor comments).

  • Response 1: Thank you for your comments. We are agreeing with your suggestion, and moved them to the main body for detailed description

Comment: Authors made significant amendments in Abstract in the revised version. Thes are reasonable and appropriate.

  • Response 2: Thank you for your suggestion. We've added the discussion based on the 3D structures of ribonucleoprotein complex (RNP), hemagglutinin (HA), neuraminidase (NA), and non - structural protein 1 (NS1) in section 5.

Comment: Authors made substantial revision in section 5. They added some discussions based on the 3D structures of ribonucleoprotein complex (RNP), hemagglutinin (HA), neuraminidase (NA), and non - structural protein 1 (NS1) . These amendments will help the readers to understand the importance of mutations introduced in each protein.

  • Response 3: Thank you so much for your comments. We made double checks throughout of the text and corrected the mistakes in revised version. We corrected the issues on the reference number and all typing mistakes after making double check in the revised version, changed “morbility” to “morbidity” online 24 page 1, replaced “Error! Reference source not found” with “Figure 1” on line 294 and 302 page 7, line 342page 8 and line 371page 9. We removed the Table 1 after considering its necessity to the overall understanding of the content. We also removed the unnecessary period on line 416 page 10.

Comment: In the revised manuscript, authors corrected minor mistakes and removed the Table 1. All these are very appropriate.

Minor comments:

  • E62k (page 10, line 415) should be E62K.
Comments on the Quality of English Language

The English is fine and does not require any improvement.